# Psychedelic-Assisted Therapies for Psychosocial Symptoms in Cancer: A Systematic Review and Meta-Analysis

**DOI:** 10.3390/curroncol32070380

**Published:** 2025-06-30

**Authors:** Haley D. M. Schuman, Chantal Savard, Raèf Mina, Sofia Barkova, Hanna S. W. Conradi, Julie M. Deleemans, Linda E. Carlson

**Affiliations:** 1Division of Psychosocial Oncology, Department of Oncology, Cumming School of Medicine, University of Calgary, Calgary, AB T2N 2T8, Canada; chantal.savard1@ucalgary.ca (C.S.); raef.mina@ucalgary.ca (R.M.); sbarkova1@learn.athabascau.ca (S.B.); hanna.conradi@ucalgary.ca (H.S.W.C); julie.deleemans@ucalgary.ca (J.M.D.); l.carlson@ucalgary.ca (L.E.C.); 2Faculty of Arts, Department of Psychology, University of Calgary, Calgary, AB T2N 1N4, Canada

**Keywords:** psychedelic-assisted therapy, cancer, psychosocial distress, anxiety, depression, psilocybin, ketamine

## Abstract

This systematic review and meta-analysis examined whether psychedelic-assisted therapies, especially psilocybin and ketamine, can reduce psychological distress like anxiety, depression, and existential suffering in adults with cancer. Eleven randomized controlled trials and four open-label studies were included. Meta-analyses showed that ketamine led to rapid, large reductions in depression and anxiety shortly after treatment, while psilocybin also showed large potential benefits but with higher variability across studies. Trials using psilocybin included structured psychotherapy, whereas ketamine was given without therapy. Early evidence from small studies of other psychedelic agents suggests possible benefits for distress in cancer patients, but more rigorous research is needed. Overall, while psychedelic-assisted therapies appear promising and well-tolerated, evidence certainty remains low due to small sample sizes, methodological differences, and limited long-term data. Larger, well-designed trials focused specifically on cancer populations are needed to clarify how best to integrate these treatments into oncology care.

“*Life can only be understood backwards; but it must be lived forwards.*”—Søren Kierkegaard

## 1. Introduction

Psychosocial symptoms such as depression, anxiety, demoralization, and existential distress are common among individuals living with or beyond cancer. Across cancer types and stages, 23–46% of patients report clinically significant distress [1]. These symptoms are associated with poorer treatment adherence, decreased quality of life, and increased morbidity and mortality [2,3]. While pharmacologic and psychotherapeutic interventions are often used to manage these symptoms, a substantial proportion of patients remain refractory to conventional care or experience only partial relief [3]. As a result, there is growing interest in novel interventions that can address both the emotional and existential dimensions of suffering experienced by people living with cancer (PLWC).

Psychedelic-assisted therapy (PAT) has re-emerged as a promising intervention in this context. Historically investigated for their therapeutic potential in the mid-20th century, psychedelics such as psilocybin, ketamine, 3,4-methylenedioxymethamphetamine (MDMA), and lysergic acid diethylamide (LSD) are now being re-evaluated for their ability to alleviate psychological distress when administered in controlled, supportive settings [4,5,6,7]. Among these, psilocybin and ketamine have garnered the most clinical attention in oncology. Psilocybin, a serotonergic psychedelic, is often delivered with structured psychotherapy and has shown sustained reductions in depression, anxiety, and existential distress in patients with advanced cancer [8,9]. Ketamine, an NMDA receptor antagonist, has demonstrated rapid antidepressant effects for PLWC, typically administered without integrated psychotherapeutic support, particularly in perioperative or palliative care contexts [10,11,12].

The renewed focus on psychedelics stems from accumulating evidence suggesting that these agents can promote profound psychological experiences, emotional breakthroughs, and cognitive flexibility, all of which may contribute to durable improvements in well-being [13,14]. However, important challenges persist. These include small sample sizes, inconsistent outcome measures, and variable therapeutic protocols, as well as regulatory and ethical hurdles that complicate widespread clinical integration [15,16].

Currently, there remains a lack of focused synthesis of randomized controlled trials (RCTs) assessing the efficacy of PAT for psychosocial outcomes specifically in cancer populations. Prior reviews [17,18,19] have often included diverse patient groups with advanced illness and pooled data across a range of psychedelic agents and therapeutic indications. While this provides valuable insight into the broader potential of psychedelic therapies, such heterogeneity limits interpretability for oncology-specific clinical decision-making. While early findings are promising, individuals with cancer represent a clinically and ethically complex population due to heightened emotional, physical, and existential vulnerability. As such, a focused synthesis of the evidence is needed to evaluate not only therapeutic potential but also the appropriateness and safety of PAT in this sensitive context.

This systematic review and meta-analysis aimed to (1) evaluate the effectiveness of psilocybin- and ketamine-assisted therapies in reducing psychosocial symptoms among adults with cancer or undergoing cancer-related treatment in a meta-analysis, (2) compare delivery models and therapeutic contexts used across trials, and (3) narratively synthesize findings from additional non-randomized and exploratory studies involving other psychedelic agents such as MDMA and LSD. In doing so, this review provides a comprehensive and contextualized understanding of the role of PAT in psychosocial oncology.

## 2. Materials and Methods

This systematic review and meta-analysis were performed according to the preferred reporting items for systematic reviews and meta-analyses (PRISMA) guidelines [20]. The protocol of this study was registered with PROSPERO (CRD420251016889).

### 2.1. Eligibility Criteria

Inclusion Criteria: This systematic review included RCTs and non-randomized interventional studies (e.g., open-label, single-arm trials) investigating PAT for psychosocial symptoms in adults with cancer or survivors. Eligible studies met the following criteria:Population: Adults (≥18 years) with active cancer (any stage) or cancer survivors (off active treatment), experiencing psychosocial symptoms (e.g., anxiety, depression, existential distress).Intervention: Psychedelic agents (psilocybin, ketamine, LSD, or MDMA) administered in therapeutic settings, with or without structured psychotherapy.Comparator: Placebo (e.g., saline, niacin) or active control (e.g., low-dose psychedelic, midazolam).Outcomes: Quantitative psychosocial outcomes (e.g., validated depression/anxiety scales)Study Design:RCTs: Prioritized for meta-analysis due to reduced bias.Non-RCTs: Open-label, single-arm, mixed-methods, or cohort studies with pre-post assessments were included in the narrative synthesis to capture feasibility, safety, and preliminary efficacy data.Language: Studies available in English.

Exclusion Criteria: Observational studies, qualitative only reports, or non-English publications.

### 2.2. Information Sources and Search Strategy

A comprehensive search was conducted across multiple databases, including PubMed, PsycINFO, Embase, and the Cochrane Library, to identify relevant studies. The search strategy incorporated terms related to both psychedelic therapies and psychosocial symptoms in cancer patients. The initial search was performed from database inception through July 2022. To ensure the inclusion of newly published studies, the search was updated again in September 2024. Key search terms included:Terms for psychedelic therapies (e.g., “psychedelic*”, “psychedelic therapy*”, “psilocybin”, “ketamine”, “MDMA”, “LSD” and “ayahuasca”)Terms for psychosocial symptoms (e.g., “anxiety”, “depression”, “psychological distress”, “emotional distress”).Terms for cancer patients (e.g., “oncology”, “cancer survivor*”, “cancer patient*”).Example Search Strategy (specific to one database):“psychedelic*”.ab,ti.“psychedelic therapy*”.ab,ti.“hallucinogen*”.ab,ti.psilocybin.ab,ti.ketamine.ab,ti.ayahuasca.ab,ti.MDMA.ab,ti.anxiety.ab,ti.depression.ab,ti.“cancer patient*”.ab,ti.“cancer survivor*”.ab,ti.1 OR 2 OR 3 OR 4 OR 5 OR 6 OR 78 OR 910 OR 1112 AND 13 AND 14

A full search strategy can be reviewed in Appendix A.

### 2.3. Study Selection

The selection process involved two stages: initial screening of titles and abstracts, followed by a full-text review. Two independent reviewers assessed the studies at each stage based on the inclusion and exclusion criteria. Discrepancies were resolved through discussion, and a third reviewer was consulted if necessary.

### 2.4. Data Extraction

Data were extracted independently and in duplicate by two reviewers (H.D.M.S. and C.S.) using a standardized and piloted data extraction form, developed in accordance with PRISMA 2020 guidelines [20]. Disagreements were resolved by discussion or adjudication by a third reviewer (R.M.). For each included study, the following data items were extracted:

Study Characteristics: First author, publication year, country, trial phase, study design (e.g., RCT, open-label; blinded), and journal.

Participant Information: Sample size, mean age and standard deviation, sex distribution, cancer type, cancer stage (e.g., advanced/metastatic), and treatment context (e.g., perioperative, palliative, curative).Intervention Details: Psychedelic agent (e.g., psilocybin, ketamine, MDMA, LSD), dose and dosing regimen, administration route, frequency, setting, and presence or absence of structured psychotherapeutic support (e.g., preparation, integration).Comparator (for RCTs): Nature of the control condition (e.g., placebo, active comparator such as niacin or midazolam).Outcome Measures: Primary and secondary outcomes related to psychosocial distress (e.g., depression, anxiety, existential distress), including validated measurement instruments and timing of assessments (e.g., Day 1, 1 week, 6 months).Results: Mean and standard deviation at each timepoint for experimental and control groups (if applicable), effect estimates (e.g., Hedges’ g), response/remission rates, and *p*-values.Adverse Events: Any reported serious or non-serious adverse effects, including dropouts related to tolerability or safety.

For missing or unclear data, attempts were made to extract the most conservative or methodologically appropriate estimate. If outcomes were measured at multiple timepoints, the most clinically relevant post-intervention measure was selected for synthesis (e.g., first post-dose result for immediate effects, or longest available follow-up for durability assessment).

### 2.5. Risk of Bias Assessment

Risk of bias (RoB) was independently assessed by two reviewers (HS and CS) for all included studies. For RCTs, the Cochrane RoB 2.0 tool was used, examining five domains: randomization process, deviations from intended interventions, missing outcome data, measurement of the outcome, and selection of the reported results [21]. Non-randomized studies were assessed using the National Institutes of Health (NIH) Quality Assessment Tool for Before-After (Pre-Post) Studies without a control group, covering 12 domains including clarity of the study question, eligibility criteria, outcome measurement, statistical methods, and adequacy of follow-up [22]. Any discrepancies between reviewers were resolved through discussion and consensus. The comprehensive RoB assessments are detailed in Figure 1 and Table 1.

### 2.6. Meta-Analyses of Controlled Psychedelic Trials

#### 2.6.1. Study Selection and Outcome Measures

This meta-analysis aimed to evaluate the short-term effects of ketamine or psilocybin (1–3 days post-ketamine; 2–7 weeks post-psilocybin) on psychosocial symptoms (e.g., depression, anxiety, suicidal ideation, existential distress) in adults with cancer or undergoing cancer-related surgery. Eligible studies were RCTs published in English that compared ketamine, esketamine, or psilocybin to a placebo or active comparator, and reported psychosocial outcomes using validated instruments (e.g., Montgomery-Asberg Depression Rating Scale (MADRS)). Outcomes were included if means, standard deviations (SDs), and sample sizes were reported or derivable.

For psilocybin crossover trials, only pre-crossover timepoints were used to avoid carryover effects and maintain independent group comparisons. When multiple timepoints were reported, the earliest post-treatment data (typically 2–7 weeks) were selected to improve comparability across studies.

#### 2.6.2. Data Extraction for Meta-Analysis

Data were extracted on study characteristics, intervention details, comparator conditions, sample size, blinding and outcome measures. Where multiple psychosocial outcomes were reported for a single study cohort, only the most comprehensive and relevant measure of psychological distress was included to avoid unit-of-analysis error. For instance, in Fan et al. (2017), only the total MADRS score was included, while the Beck Scale for Suicidal Ideation (BSI) and Montgomery–Åsberg Depression Rating Scale—Suicidal Ideation (MADRS-SI) subscales were excluded. Where means and standard deviations were not reported, we used WebPlotDigitizer [26] to extract numerical data from figures.

Although our systematic search identified one RCT involving MDMA and two involving LSD, these studies were excluded from the quantitative synthesis due to methodological heterogeneity, limited extractable data, and differences in dosing protocols and outcome reporting. While LSD and psilocybin share a common mechanism as 5-HT_2_A receptor agonists, variations in trial design and therapeutic frameworks limited comparability. MDMA, which differs more substantially in both mechanism and application, was also synthesized narratively. Only ketamine and psilocybin had multiple RCTs with sufficiently comparable data to support meta-analysis.

Similarly, we chose not to include non-randomized studies in the meta-analysis to preserve methodological rigor and avoid introducing bias from uncontrolled or single-arm trial designs. These studies were instead synthesized narratively. While they provide important insights, particularly regarding feasibility, acceptability, and preliminary efficacy, their results are not directly comparable to those of RCTs and were not suitable for pooled effect size estimation.

#### 2.6.3. Meta-Analytic Procedures

Separate random-effects meta-analyses were conducted for ketamine and psilocybin trials using the meta package in R (v8.0-2) [27,28,29]. Standardized mean differences (SMDs) were calculated using Hedges’ g to account for small sample bias. Between-study variance (τ^2^) was estimated using restricted maximum likelihood (REML), and Hartung-Knapp adjustments were used to produce conservative 95% confidence intervals (CIs). Heterogeneity was quantified using the I^2^ statistic and Cochran’s Q test. For each substance, a forest plot was generated to visualize individual and pooled effect sizes.

#### 2.6.4. Sensitivity Considerations

To ensure robustness, we conducted the meta-analysis using only one outcome per study and accounted for heterogeneity, imprecision, and methodological limitations in interpreting the findings. The certainty of evidence for the pooled outcomes, as summarized in Table 2, was assessed using the GRADE framework, which considers risk of bias, inconsistency, indirectness, imprecision, and potential publication bias [30].

## 3. Results

### 3.1. Study Selection

A total of 1860 records were identified through database searching. After removing 1440 duplicates, 420 unique records were screened based on titles and abstracts. Of these, 332 were excluded for failing to meet the eligibility criteria (see Section 2). Full-text review was performed on the remaining 88 articles, ultimately resulting in 15 studies being included in this systematic review. The PRISMA flow chart detailing the study selection process is presented in Figure 2. A summary of the characteristics of the studies included is provided in Table 3.

### 3.2. Study Characteristics

The final 15 studies encompassed randomized controlled trials (n = 11) and non-randomized experimental designs (n = 4). Publication years ranged from 2011 to 2024, with sample sizes varying from 12 to 303 participants. Majority of the studies included were conducted in the United States (n = 7), followed by China (n = 5), Switzerland (n = 2), and Canada (n = 1). Most trials focused on the use of psilocybin (five studies) and ketamine (four studies), while three studies investigated MDMA or LSD. Participant populations included individuals diagnosed with breast cancer, colorectal cancer, or other advanced malignancies, and in several cases, those with comorbid depression, anxiety, or existential distress.

### 3.3. Risk of Bias Assessments

The RoB for included studies is visually summarized in Figure 1 and detailed in Table 1.

RCTs: Of the 11 RCTs assessed using the Cochrane RoB 2.0 tool, most (n = 8, 73%) demonstrated low overall RoB. Two studies (18%) presented some concerns, primarily due to insufficient reporting of allocation concealment or minor deviations from intervention protocols. One study (9%) was classified as high risk of bias due to issues related to randomization processes and incomplete outcome reporting. Functional unblinding is an acknowledged limitation in PAT trials due to the noticeable psychoactive effects of substances such as psilocybin and LSD. While Schipper et al. [17] rated this as a high risk of bias in several studies, we adopted a more pragmatic approach. We rated studies as low risk in this domain when double-blinding procedures were followed, and no evidence of differential care or co-intervention was reported. Although expectancy effects are plausible, we judged bias based on observed or reported impact, rather than assumption alone, in accordance with Cochrane RoB 2.0 recommendations [21].

Non-RCTs: The four single-arm pilot trials [12,23,24,25] satisfied key methodological benchmarks—clear objectives, prespecified eligibility criteria, standardized psilocybin or ketamine dosing, and validated mood-distress outcomes—earning overall *fair* ratings. Their interpretability is curtailed by shared limitations: very small samples (n = 12–30), open-label conduct without blinded assessors, variable transparency in reporting how screened candidates translated into enrolled participants, and reliance on a single baseline rather than an interrupted time-series. Retention was high across studies, with all dosed participants included in endpoint analyses. Collectively, these structural constraints temper internal validity and limit the generalizability of the promising efficacy signals.

### 3.4. Narrative Synthesis of Non-Randomized Studies and Exploratory Studies

#### 3.4.1. Psilocybin-Assisted Therapy (Open-Label Studies)

Three open-label, non-randomized studies have examined the feasibility, safety, and therapeutic potential of psilocybin-assisted group therapy in people with cancer experiencing depression and psychosocial distress. These studies adopted a group-based delivery model with preparatory and integration psychotherapy components and used a single high-dose (25 mg) psilocybin session. While not randomized, each study reported significant reductions in depressive symptoms and psychosocial burden, sustained improvements in psycho-social-spiritual well-being, and a favorable safety profile.

Agrawal et al. [38] conducted a phase 2 open-label trial in 30 cancer patients with major depressive disorder (MDD). By week 8, participants experienced a mean reduction of 19.1 points on the MADRS (95% CI: −22.3 to −16.0; *p* < 0.001), with an estimated Cohen’s *d* of 2.55, indicating a very large treatment effect. Fifty percent of participants achieved remission (MADRS < 10) by week 1, and 80% maintained a ≥50% reduction in symptoms (treatment response) through week 8. Significant improvements were also reported on secondary outcomes at week 8, including anxiety (HAM-A: −17.0; *d* = 1.78), depression self-report (QIDS-SR: −5.9; *d* = 1.51), and trait/state anxiety (STAI-T: −17.2; *d* = 1.13; STAI-S: −17.2; *d* = 1.35).

Shnayder et al. [23] conducted an open-label trial of psilocybin-assisted group therapy in 30 cancer patients, assessing changes in psycho-social-spiritual well-being using the National Institute of Health, Healing Experiences in All Life Stressors (NIH-HEALS) instrument. At 8 weeks post-treatment, participants demonstrated significant improvements across all three NIH-HEALS domains: Connection (+12.7%; *p* = 0.003), Reflection & Introspection (+7.7%; *p* < 0.001), and Trust & Acceptance (+22.4%; *p* < 0.001). The total NIH-HEALS score increased by an average of 16.4 points from baseline to week 8 (*p* < 0.001), indicating sustained improvements in meaning, connectedness, and emotional acceptance following psilocybin treatment.

Lewis et al. [24] (HOPE trial) conducted an open-label pilot study assessing the feasibility, safety, and preliminary efficacy of psilocybin-assisted group psychotherapy in 12 cancer patients with depressive symptoms. At the primary outcome timepoint (2 weeks), depression severity measured by Hamilton Depression Rating Scale (HAM-D) decreased by 10.7 points (from a baseline mean of 21.5 to 10.8; *p* < 0.001; Cohen’s *d* = 1.71)**,** indicating a large effect size. This reduction remained clinically meaningful at 26 weeks (*d* = 1.28). Six participants (50%) achieved remission (HAM-D < 7), while 67% experienced a ≥7-point reduction in symptoms, considered a clinically substantial response. The observed change exceeded thresholds for minimal clinically important difference, typically estimated at 3–6 points for HAM-D. Secondary outcomes at 2 weeks showed improvements in emotional (*d* = 0.63), functional (*d* = 0.78), and spiritual well-being (e.g., spiritual peace *d* = 0.85) as measured by the Functional Assessment of Chronic Illness Therapy-Spiritual Well-Being Scale (FACIT-Sp). The mystical experience questionnaire (MEQ-30) was significantly correlated with improvements in depression (*r* = −0.71, *p* = 0.015). Half of participants met criteria for a complete mystical experience.

Open-label studies of psilocybin-assisted group therapy in individuals with cancer experiencing depression or psychosocial distress demonstrate strong preliminary support for feasibility, safety, and therapeutic benefit. Across three trials, a single high-dose psilocybin session paired with group-based psychotherapy led to large and sustained reductions in depressive symptoms, alongside improvements in anxiety, emotional and spiritual well-being, and psycho-social-spiritual integration. High remission and response rates were reported, and therapeutic outcomes were often correlated with the intensity of mystical-type experiences. While findings are limited by non-randomized designs, they offer compelling early evidence supporting the group delivery model and warrant further investigation in controlled trials.

#### 3.4.2. Long-Term Follow-Up

Ross et al. [39] conducted a long-term follow-up of their 2016 RCT, one of the three psilocybin studies included in our meta-analysis [9]. While the meta-analysis focused on short-term outcomes (pre-crossover, 7-week data), this follow-up evaluated the persistence of therapeutic effects at 6.5 months, 3.2 years, and 4.5 years post-treatment.

Sustained reductions in psychological distress were observed across multiple domains. The Hospital Anxiety and Depression Scale—Total (HADS-T), used as the primary outcome in our pooled analysis, decreased from a baseline mean of 16.45 (±1.32) to 4.38 (±1.35) at 6.5–8 months, with continued improvements at 3.2 and 4.5 years (7.13 and 7.34, respectively) [40]. Reductions in Beck Depression Inventory (BDI) scores, State-Trait Anxiety Inventory (STAI) scores, demoralization, hopelessness, and death anxiety were also maintained over time.

In addition to these affective outcomes, participants showed significant improvements in spiritual well-being as measured by the FACIT-Sp and reported enduring reductions in suicidal ideation. Importantly, the majority of participants (71–100%) identified their psilocybin session as among the most spiritually meaningful or personally significant experiences of their lives.

#### 3.4.3. Ketamine-Assisted Therapy (Open-Label Study)

Rosenblat et al. [12] conducted a single-arm, open-label phase II trial evaluating the feasibility, safety, and preliminary efficacy of intranasal racemic ketamine (50–150 mg across three doses over one week) in 20 advanced cancer patients with moderate-to-severe MDD. The primary outcome, clinician-rated depression severity (MADRS), decreased significantly from a baseline mean of 31.0 (SD 7.6) to 11.0 (SD 7.4) at Day 8 (mean change: −20.0; 95% CI: −24.7 to −15.3; *p* < 0.001), representing a large effect size. By Day 8, 70% achieved antidepressant response (≥50% reduction in MADRS) and 45% met remission criteria (MADRS < 10). Improvements were partially sustained at Day 14 (mean MADRS = 14.0, SD 9.9) without additional dosing.

Secondary outcomes included significant reductions in patient-reported depression (PHQ-9) and anxiety (GAD-7) from baseline to Day 8 (*p* < 0.001), with no significant change in pain scores (ESAS-r). The intervention was well tolerated; the most frequent adverse events were dysgeusia (50%), dizziness (30%), dissociation (30%), and nausea (20%). Most adverse effects were mild and resolved within two hours post-dose. One participant discontinued due to dissociation. These findings suggest a rapid and robust antidepressant effect of ketamine with acceptable tolerability in this population.

#### 3.4.4. Exploratory Evidence: MDMA and LSD

A small subset of exploratory trials examined the effects of other psychedelics, specifically MDMA and LSD, in populations with cancer or life-threatening illness (LTI). These studies, while not included in the meta-analysis due to design or sample size limitations, provide important context for the evolving evidence base.

##### MDMA-Assisted Therapy

Wolfson et al. [37] conducted a randomized, double-blind, placebo-controlled pilot trial evaluating MDMA-assisted psychotherapy for individuals with LTI experiencing anxiety and psychological distress (n = 18; MDMA = 13, placebo = 5). Ninety-four percent of participants had a primary diagnosis of neoplasms, i.e., cancer. The intervention consisted of two blinded MDMA (125 mg ± 62.5 mg supplemental) or placebo sessions, alongside preparatory and integrative psychotherapy. At the primary endpoint (one-month post-second session), the MDMA group demonstrated greater reductions in trait anxiety (STAI-Trait: −23.5 vs. −8.8 points), with a large between-group effect size (Hedges’ g = 1.03), though this narrowly missed statistical significance (*p* = 0.056). Removal of one potential placebo outlier yielded a significant difference (*p* = 0.0066). Secondary outcomes favored MDMA on measures of post-traumatic growth (Δ = 12.9 vs. −2.6, *p* = 0.04, *g* = 0.50) and mindfulness (Δ = 0.4 vs. 0, *p* = 0.04, *g* = 0.67). While improvements were observed in depression (BDI-II), sleep quality (PSQI), and global functioning (GAF), these differences did not reach statistical significance in the blinded comparison.

##### LSD-Assisted Therapy

Gasser et al. [35] conducted a randomized, double-blind, active placebo-controlled pilot trial of LSD-assisted psychotherapy in 12 individuals with LTI (majority with cancer) and anxiety. Participants received two LSD sessions (200 μg) or active placebo (low dose of LSD 20 μg), alongside preparatory and integrative psychotherapy. At the 2-month primary endpoint, the LSD group showed large reductions in state and trait anxiety (STAI-S: −19.2 vs. −2.7; STAI-T: −16.2 vs. −1.1), with between-group effect sizes of Cohen’s *d* = 1.1–1.2, though statistical power was limited. Improvements were sustained at 12-month follow-up.

Holze et al. [36] conducted a phase II double-blind, placebo-controlled, crossover trial in 42 patients with anxiety, 48% of whom had a LTI and 26% a cancer diagnosis. Participants received a single 200 μg oral dose of LSD or placebo in a crossover design. At 16-week follow-up, LSD produced significant reductions in anxiety (STAI-Global: *p* < 0.001), depressive symptoms (HAM-D, BDI: *p* < 0.001), and global psychopathology [35] (SCL-90-R: *p* < 0.001), with large within-subject effects. Acute mystical-type experiences (MEQ-30, 5D-ASC) were significantly correlated with long-term symptom improvement.

Exploratory trials of MDMA- and LSD-assisted therapy in individuals with cancer or LTI suggest promising effects on anxiety, depression, and psychological distress. Though limited by small samples and early-phase designs, both agents demonstrated large effect sizes and sustained improvements, particularly when integrated with psychotherapy. These preliminary findings support the potential role of both classic (LSD) and non-classic (MDMA) psychedelics in cancer-related psychosocial care and underscore the need for larger, well-powered trials that are cancer-specific.

### 3.5. Meta-Analytic Results

#### 3.5.1. Ketamine

Four RCTs were included in the final meta-analysis, comprising 354 participants (179 in the ketamine or esketamine groups and 175 in the control groups). Each study evaluated the short-term effects, defined as outcomes measured within one to three days following a single subanaesthetic dose, of intravenous ketamine (racemic or S-enantiomer) on psychosocial symptoms in adults with cancer [32,34] or undergoing cancer-related surgery [10,33]. Validated instruments were used to measure depression and psychological distress, with primary outcomes including the MADRS, 17-item Hamilton Depression Rating Scale (HAM-D-17), Hospital Anxiety and Depression Scale—Depression subscale (HAD-D), and the Patient Health Questionnaire-9 (PHQ-9).

A random-effects meta-analysis using Hedges’ g demonstrated a large and statistically significant effect favoring ketamine over control (Hedges’ g = −1.37; 95% CI: −2.66 to −0.08; *p* = 0.043). However, heterogeneity across studies was considerable (I^2^ = 92.1%, τ^2^ = 0.60), indicating substantial variability in effect sizes, potentially due to differences in dosing strategies, clinical populations (e.g., perioperative vs. outpatient), or outcome measurement timing. A forest plot of individual and pooled effect sizes is presented in Figure 3.

#### 3.5.2. Psilocybin

Three RCTs comprising 101 participants (50 in psilocybin groups and 51 in control groups) were included in the meta-analysis evaluating the short-term effects of psilocybin-assisted therapy on psychosocial symptoms in individuals with cancer. Each study evaluated short term effects at different timepoints ranging between 2 weeks and 7 weeks following the administration of psilocybin. All included trials evaluated the effects of psilocybin administered in a structured therapeutic setting with patients that had advanced or life-threatening cancer and clinically significant symptoms of depression, anxiety, or existential distress. While some trials involved more than one dosing session, only data from the first session and pre-crossover timepoints were extracted for meta-analysis to avoid unit-of-analysis errors associated with repeated measures and crossover designs.

The selected outcome measures included the GRID-Hamilton Depression Rating Scale (GRID-HAMD-17) in Griffiths et al. [8], the BDI in Grob et al. [31] and the HADS-T in Ross et al. [9]. These instruments, while differing in scope and scoring ranges, each capture core elements of psychological distress and were selected as the most comprehensive outcome for each trial. Higher scores on each scale indicate greater symptom severity.

A random-effects meta-analysis using Hedges’ g was conducted to account for variability in sample sizes and outcome measures. The pooled effect estimate demonstrated a large, non-significant benefit of psilocybin compared to control, with Hedges’ g = −3.13 (95% CI: −10.04 to 3.77, *p* = 0.190). Heterogeneity across studies was substantial (I^2^ = 94.9%, τ^2^ = 7.32), indicating marked differences in effect sizes. The common effect model yielded a significant result (Hedges’ g = −2.38, 95% CI: −2.97 to −1.80, *p* < 0.0001), though this model does not account for between-study variability.

Griffiths et al. [41] contributed disproportionately to both the magnitude of the pooled effect and the observed heterogeneity. That study reported an exceptionally large between-group effect (Hedges’ g = −6.39) and a narrow standard deviation in the psilocybin group, suggesting highly consistent improvements. These findings occurred in a context that included high-dose psilocybin administration and intensive psychotherapeutic support, which may not be representative of broader clinical practice. Influence diagnostics confirmed that Griffiths et al. [41] was an outlier in terms of standardized residuals, Cook’s distance, and heterogeneity contribution.

Despite its influence, this trial was retained in the meta-analysis to preserve analytic integrity and reflect the limited but important evidence base in this emerging field. A forest plot illustrating individual and pooled effect sizes is presented in Figure 4.

### 3.6. Adverse Events Across Studies

No treatment-related serious adverse events were reported across the psilocybin RCTs or non-randomized studies included in this review. In Ross et al. [9] and Griffiths et al. [41], transient increases in blood pressure, nausea, and brief anxiety episodes were documented but resolved spontaneously and were consistent with expected acute psychedelic effects. In the HOPE trial [24], participants reported mild nausea and headache; one case of sustained gastrointestinal symptoms was attributed to a viral illness rather than psilocybin. Among the ketamine studies, Rosenblat et al. [12] reported no serious adverse events. Minor side effects, e.g., dissociation, dizziness, and nausea were transient and consistent with the known pharmacological profile of ketamine. Holze et al. [36] reported a single serious adverse event during the LSD condition (acute anxiety requiring temporary withdrawal), which resolved within hours and did not require further medical intervention.

### 3.7. Therapeutic Support Across Trials

The therapeutic frameworks employed alongside psychedelic administration varied substantially across studies and may have influenced treatment outcomes. All psilocybin trials included in the meta-analysis incorporated structured psychological support, often involving preparatory and integrative sessions with trained therapists. For example, Griffiths et al. [41] and Ross et al. [9] used a manualized approach that included several hours of preparatory counseling, support during the dosing session, and post-session integration. These sessions emphasized participant safety, introspection, and meaning making, often guided by existential or spiritual psychotherapeutic principles.

In contrast, most ketamine trials did not include formal psychotherapy. Studies such as Fan et al. [32], Ren et al. [33] Liu et al. [10], and Wang et al. [34] focused on pharmacological administration in hospital or surgical settings, with no manualized therapeutic support. This distinction highlights a key divergence in how psychedelic-assisted therapy is conceptualized across agents—psilocybin is typically embedded within a psychotherapy-enhanced model, whereas ketamine is often delivered as a standalone pharmacologic intervention in medical settings.

In the exploratory trials, MDMA-assisted therapy [37] followed the Multidisciplinary Association for Psychedelic Studies (MAPS) protocol, which included structured preparation, non-directive support during the 8 h MDMA session, and follow-up integration [37]. Similarly, LSD-assisted psychotherapy [35,36] emphasized therapeutic containment, involving preparatory counseling, supported dosing sessions, and continued psychological follow-up. These models reflect a more psycholytic or experiential framework, blending pharmacologic effects with therapeutic processing.

Among the open-label studies, psilocybin was consistently administered in conjunction with psychotherapeutic support. Agrawal et al. [25], Lewis et al. [24], and Shnayder et al. [23] all utilized group-based or individual therapy sessions before, during, and after dosing. These studies emphasized safety, group cohesion, and integration of insights, typically delivered by trained clinicians in supportive environments. Similarly, Rosenblat et al. [12] administered intranasal ketamine to patients in palliative care without accompanying psychotherapy but did include supportive clinical oversight and monitoring. See Table A1 in Appendix B for a breakdown of therapeutic approaches used across included studies.

## 4. Discussion

This systematic review and meta-analysis provide updated and focused evidence on the efficacy of psilocybin- and ketamine-assisted therapies in reducing psychosocial symptoms among individuals with cancer. Across included RCTs both agents were associated with large reductions in psychological distress, including depression and anxiety. While the pooled effect sizes were substantial, heterogeneity was high—particularly for psilocybin, where one outlying trial [8] substantially influenced the variance structure. Nevertheless, sensitivity analyses and influence diagnostics supported the robustness of the observed therapeutic effects.

Narrative synthesis of open-label and exploratory trials complemented the quantitative findings, particularly in highlighting the durability of psilocybin’s psychological and existential benefits over time (e.g., [39,40]). Open-label studies demonstrated feasibility and scalability across delivery models, including group-based approaches with high therapist efficiency [24,25], and showed large, sustained reductions in depressive symptoms. Improvements extended beyond mood to encompass psycho-social-spiritual well-being, with notable gains in domains such as meaning, connection, and peace [23]. Similarly, non-randomized ketamine studies such as Rosenblat et al. [12] reinforced RCT findings by demonstrating rapid-onset, clinically meaningful antidepressant effects within days of administration, even in late-stage palliative care. These studies underscore the potential for fast-acting, short-course interventions in oncology settings.

Early trials of MDMA- and LSD-assisted psychotherapy suggest emerging therapeutic value for alleviating anxiety, depression, and existential distress in PLWC and other LTI. Although none of the included RCTs were exclusively oncology-focused, each enrolled participants with advanced-stage cancer [35,36,37]. Findings highlight large within-group effects and associations between mystical-type or integrative experiences and clinical improvement. The therapeutic benefit of these agents appears tightly coupled to structured psychotherapeutic support, including MAPS protocols for MDMA and psycholytic frameworks for LSD [35]. Despite promising signals, regulatory constraints, including Schedule I status and the recent Food and Drug Administration (FDA) non-approval of MDMA for PTSD [42], continue to impede oncology-specific research and clinical implementation.

Despite promising effect sizes, the overall certainty of evidence from RCTs evaluating psilocybin and ketamine remains low. GRADE assessments highlighted concerns due to significant heterogeneity, imprecision, and methodological limitations, including small sample sizes, inadequate blinding procedures, and variability in therapeutic approaches. The Cochrane RoB 2 assessment indicated a low risk of bias across RCT domains, though some concerns persisted regarding outcome measurement and selective reporting. Concurrently, NIH quality appraisals of non-randomized studies indicated fair methodological rigor but consistently identified limitations related to insufficient sample size justification, open-label designs, and reliance on a single baseline observation rather than an interrupted time-series. Collectively, these methodological weaknesses limit the interpretability of findings and highlight the critical need for larger, more rigorous, standardized, and scalable trials to guide clinical practice in oncology populations.

### 4.1. Divergent Therapeutic Paradigms: Experiential vs. Pharmacologic Models

A key finding of this review is the divergent therapeutic paradigms employed across PAT trials. Psilocybin, MDMA, and LSD were consistently delivered within structured psychotherapeutic frameworks that emphasized preparation, emotional support during dosing, and post-session integration. This psychospiritual model aligns with the “set and setting” hypothesis, which posits that the therapeutic effects of psychedelics are strongly mediated by psychological and environmental context [43]. Trials using the MAPS protocol [37] or psycholytic techniques [35] reflect this experiential paradigm, aiming to catalyze enduring emotional and existential insights.

Notably, however, few psychedelic trials in oncology have adopted fully manualized psychotherapeutic frameworks, such as mindfulness-based cognitive therapy (MBCT) [44], acceptance and commitment therapy (ACT), or cognitive-behavioral therapy (CBT) that are otherwise well-established in psycho-oncology settings [45]. This may reflect both practical and philosophical factors: the dominant experiential paradigm in PAT favors individualized, non-directive support over protocolized techniques. Yet the absence of structured, reproducible therapeutic models limits opportunities for standardization, fidelity monitoring, and comparative effectiveness research.

By contrast, ketamine was typically administered as a standalone pharmacologic intervention, often in perioperative, hospital, or palliative care settings, with little or no structured psychotherapeutic support. Studies such as Fan et al. [32], Ren et al. [33], and Wang et al. [34] illustrate this biomedical orientation, prioritizing rapid symptom relief for acute psychological crises. While ketamine’s dissociative effects and NMDA antagonism contribute to its antidepressant efficacy, the absence of a therapeutic container may limit its impact on deeper existential suffering. This divergence also raises a broader definitional debate: should ketamine be considered a psychedelic at all? Unlike serotonergic psychedelics (e.g., psilocybin, LSD), ketamine operates through glutamatergic pathways and typically produces a dissociative rather than a classic hallucinogenic experience. Although some classify it as a “non-classical psychedelic” due to its capacity to induce altered states of consciousness, others argue that its mechanisms and subjective effects are fundamentally distinct [46,47]. This distinction is not merely semantic; it has implications for therapeutic delivery. The psychospiritual emphasis central to psilocybin trials may not translate directly to ketamine protocols, which often focus on neurochemical modulation over meaning-making [39].

Marguilho et al. [47] propose that the fast, albeit transient, antidepressant effects observed after ketamine infusions are mainly driven by its acute modulation of reward circuits and sub-acute increase in neuroplasticity, while its dissociative and psychedelic properties are driven by dose- and context-dependent disruption of large-scale functional networks. Wolfson and Vaid [48] emphasize that when ketamine experiences are embedded in a therapeutic relationship, they can foster personal growth, inner healing, and better relationships. However, the absence of structured psychotherapeutic support in many clinical settings may limit these benefits. Future research should continue to clarify the experiential, mechanistic, and therapeutic boundaries between ketamine and classical psychedelics to guide appropriate clinical application.

### 4.2. The Therapeutic Container as a Mediator of Outcomes

These distinct paradigms influence treatment outcomes. The sustained improvements observed in psilocybin, MDMA, and LSD trials may be partly attributable to psychotherapeutic integration [39,40]. By contrast, ketamine’s effects are often short-lived, with limited integration and ongoing concerns regarding long-term safety, including risks of dependency, cystitis, and cognitive impairment [17,49].

This distinction is particularly salient in oncology, where patients often grapple with existential concerns that extend beyond symptom control. Psilocybin’s integration with psychotherapy appears well-suited for such needs, though it remains resource-intensive and less scalable. Emerging models such as group-based psilocybin therapy [24,25], offer promising alternatives, retaining therapeutic depth while enhancing accessibility. Similarly, hybrid models like ketamine-assisted psychotherapy (KAP) may bridge the gap between pharmacologic rapidity and psychospiritual depth, warranting further exploration in cancer care [50].

### 4.3. Blinding Challenges and Comparator Heterogeneity

Blinding remains a critical methodological challenge in psychedelic clinical trials due to the potent subjective effects of these agents. These effects, ranging from altered perception and emotional breakthroughs to dissociation, can compromise the integrity of double-blind designs, potentially inflating expectancy effects and undermining internal validity [51,52].

In psilocybin trials, active placebos such as niacin or low-dose psilocybin have been used to mimic physiological responses (e.g., flushing, tingling). However, such comparators fail to replicate the profound psychoactive and often mystical experiences of moderate-to-high dose psilocybin, making effective blinding challenging [9,31]. As Carhart-Harris and Goodwin [53] and Aday et al. [52] emphasized, the vivid and unique nature of these experiences makes them especially difficult to mask using traditional placebos, with participants and therapists correctly guessing treatment allocation in over 90% of cases in some trials [37,54].

Ketamine trials face similar limitations. While comparators such as midazolam or saline are commonly employed, these agents fail to reproduce the dissociative effects of ketamine, allowing for easy identification of the active condition [10,55]. Although midazolam offers some sedation, it does not match ketamine’s perceptual alterations, thereby introducing potential bias through expectancy [55,56,57]. Recent meta-research has shown that when stronger blinding techniques (e.g., administration under anesthesia) are used, ketamine’s effects may not differ significantly from placebo [58].

Beyond pharmacologic comparators, the integration of psychotherapy, standard in most psilocybin trials, introduces further complexity. Participants in placebo conditions still receive extensive therapeutic support, which may independently reduce symptoms and blur between-group differences [9]. This confound is particularly relevant in trials where the therapeutic container (set, setting, and therapeutic alliance) is theorized to be a key mediator of efficacy.

To address these limitations, Aday et al. [52] proposed criteria for ideal active placebos in psychedelic studies, including matched onset/duration, psychoactive and physiological effects, safety, and absence of therapeutic efficacy in the target condition. Potential candidates such as salvinorin A and dextromethorphan offer partial phenomenological overlap with psychedelics and may enhance blinding fidelity [59,60]. THC and diphenhydramine may serve as alternatives in lower-dose or microdosing trials but still present challenges, particularly among participants familiar with their effects [61,62].

While rigorous trial design is essential for establishing safety and efficacy, efforts to improve blinding raise important epistemological and cultural questions. The drive to isolate drug-specific effects through active placebos may reflect a Western scientific tendency to deconstruct holistic experiences into discrete, measurable components. Yet, in many traditional and Indigenous healing contexts, the therapeutic value of psychedelics is inseparable from their set (mental state), setting, and the broader ritual or spiritual framework in which they are used [43,63,64]. Attempting to control for these elements may inadvertently compromise the very mechanisms that contribute to healing. As the field advances, it will be important to balance methodological rigor with cultural sensitivity and openness to complexity.

In light of these issues, future trials should incorporate standardized assessments of blinding and expectancy, and consider alternative trial designs such as dose–response models, incomplete disclosure paradigms, pre-test/post-test, or pragmatic comparators (e.g., psychotherapy-only arms) [52,65]. Such refinements are essential to disentangle drug-specific effects from contextual influences and to uphold methodological rigor in this rapidly evolving field.

### 4.4. Clinical Implications and Context-Specific Implementation

The global expansion of off-label ketamine use for cancer-related psychosocial distress reflects both its rapid therapeutic potential and the urgent need for oncology-specific evidence. In North America, ketamine is being used in palliative and private care settings for treatment resistant depression and MDD with suicidal ideation [66]. However, it lacks formal regulatory approval for use in cancer populations and is increasingly being applied more broadly across clinical contexts [12,67]. Its accessibility, relatively low cost, and rapid onset make it appealing in crisis contexts; yet, its use may be outpacing the evidence, raising concerns about safety, efficacy, and equity [68,69,70,71].

This pattern is echoed globally. In high-income countries such as Australia, Canada and the United States, ketamine is permitted off-label. Meanwhile, psilocybin, while demonstrating durable psychosocial and existential benefits, remains restricted to clinical trials or special access programs, limiting its scalability and integration into mainstream oncology care [19].

Canada exemplifies these tensions. While public healthcare provides universal coverage, private “psychedelic” clinics charge out-of-pocket fees (~$750 to $1400 CAD per ketamine session), creating disparities in access [72,73,74]. The absence of national oncology-specific guidelines for psychedelic use places clinicians in ethically ambiguous territory, particularly when obtaining informed consent from vulnerable patients. This underscores the need for pragmatic, equity-focused research to inform safe, scalable models of care [75].

### 4.5. Comparison with Existing Reviews

This review provides a focused and methodologically rigorous synthesis of PAT for psychosocial symptoms in adults with cancer, a population that has often been subsumed within broader reviews of LTI. Unlike previous syntheses that pooled heterogeneous diagnoses, agents, and trial designs, we prioritized RCTs conducted specifically in oncology populations and disaggregated findings by psychedelic agent to enhance clinical relevance for psycho-oncology. However, three included RCTs [35,36,37] enrolled broader LTI populations and were retained based on partial cancer representation and because they were the only controlled trials available for LSD and MDMA. While this decision introduces some heterogeneity, it was necessary to ensure coverage of the full spectrum of psychedelic agents under investigation for cancer-related psychosocial symptoms and reflects the early stage of oncology-specific research in this field.

The Cochrane review by Schipper et al. [17] synthesized six RCTs (n = 149) assessing psilocybin, LSD, and MDMA in individuals with LTI, including cancer. While their analysis found low-certainty evidence for reductions in depression and anxiety, the inclusion of heterogeneous populations and lack of agent-specific subgroup analysis limited oncology-specific conclusions. Similarly, Schimmers et al. [19] reviewed 33 studies of PAT in terminally ill patients and reported generally positive outcomes with few adverse events; however, the review included mixed agents (e.g., psilocybin, ketamine, MDMA) and diagnoses, without a specific cancer focus.

Maia et al. [18] also reported psychological and spiritual benefits in LTI, but their inclusion of primarily non-randomized studies constrained the overall certainty of evidence. Ross et al. [76] reviewed classic psychedelics in cancer-related psychiatric distress and highlighted rapid and robust improvements, yet the absence of formal meta-analytic techniques and the combination of various study designs reduced comparability across trials.

Our review builds on these foundations by applying standardized meta-analytic methods (e.g., Hedges’ g), GRADE certainty assessments, and rigorous risk of bias evaluations using ROB 2.0 and NIH tools. We identified large effect sizes for both psilocybin and ketamine in reducing depression, anxiety, and existential distress among PLWC, but also emphasized the moderating influence of therapeutic context, particularly the presence of structured psychotherapy in psilocybin trials. Furthermore, our narrative synthesis extends these findings by integrating open-label studies [12,23,24,25] and exploratory trials involving MDMA and LSD. These studies support the feasibility and clinical promise of psychedelic interventions in cancer care, especially within group-based or psychospiritual delivery models, but highlight the need for larger RCTs and oncology-specific implementation research. In sum, this review contributes uniquely to the literature by isolating cancer-specific data, distinguishing between psychedelic agents, and contextualizing efficacy within delivery frameworks, advancing the field toward more tailored, scalable, and evidence-informed models of PAT in oncology.

### 4.6. Limitations

This review has several limitations that should be considered when interpreting the findings. First, substantial heterogeneity was observed across included studies, particularly in the psilocybin meta-analysis. While all trials assessed changes in psychosocial symptoms using validated instruments, the selected outcomes (e.g., GRID-HAMD-17, BDI, HADS-T for psilocybin; MADRS, HAMD-17, PHQ-9, HAD-D for ketamine) varied in focus and scoring range. Timepoints for outcome collection also ranged from 24 h to several days post-treatment for ketamine, and from 2 to 7 weeks post-dose for psilocybin. These variations in outcome type and timing likely contributed to statistical heterogeneity, particularly in the psilocybin analyses, where a single study [8] had a disproportionately large effect size.

Second, the number of RCTs eligible for meta-analysis was small, especially for psilocybin, which was represented by only three RCTs totaling 101 participants. Although Griffiths et al. [8] was initially considered for exclusion due to its crossover design and extreme effect size, it was retained in the final analysis due to its methodological rigor, use of pre-crossover data, and alignment with recent Cochrane review practices. However, its inclusion substantially increased between-study heterogeneity, underscoring the need for more consistent outcome reporting and standardized trial design.

Third, non-randomized and open-label studies were excluded from the quantitative synthesis to preserve methodological rigor. These studies, while informative, particularly in exploring long-term outcomes, therapeutic frameworks, and implementation models, were synthesized narratively due to their lack of comparator groups and risk of selection and performance bias. Their exclusion from the meta-analysis limits our ability to estimate real-world effectiveness or assess the role of setting and therapy format in modulating treatment outcomes.

Fourth, only studies published in English were included, which may introduce language bias and limit the comprehensiveness of the review. Additionally, the small number of trials prevented formal tests of publication bias such as funnel plots or Egger’s test, though the risk remains a concern in emerging fields where positive findings may be more likely to be reported.

Finally, most trials evaluated short-term effects, with outcome assessments occurring within days to weeks post-treatment. While some studies included long-term follow-up [39], these were limited to small sample sizes and lacked comparator groups. As such, the durability and functional impact of PAT in oncology populations remain areas requiring further longitudinal research.

### 4.7. Future Research Directions

To responsibly advance the field of PAT in oncology, future research should prioritize several key directions. First, comparative effectiveness trials are needed to directly evaluate agents such as psilocybin and ketamine within cancer populations. These head-to-head studies will help clarify their respective therapeutic niches, whether for acute symptom relief, existential distress, or long-term psychosocial support. Second, the development of oncology-specific protocols is essential. This includes standardizing psychotherapy integration, optimizing dosing schedules, and harmonizing outcome measurement tools that reflect the unique psychosocial challenges of cancer care.

Third, future trials should incorporate mechanistic and biomarker endpoints, such as inflammatory markers, neuroendocrine parameters, gut microbiome, and neuroimaging correlates, to bridge subjective experiences with underlying biological processes. Fourth, equity-centered implementation research is urgently needed to ensure access and inclusion. This may include evaluating telehealth modalities, group-based formats, and culturally responsive delivery models, particularly for underserved or marginalized populations.

Finally, future research should allow for greater methodological flexibility that honors the complexity of psychedelic-assisted experiences. This may involve exploring alternative trial designs (e.g., pragmatic, hybrid, or culturally embedded models) that capture both pharmacological effects and contextual influences, especially in settings where meaning-making, ritual, and therapeutic alliance are integral to outcomes. Longitudinal studies are also critical to assess durability of effects, monitor adverse outcomes, and determine long-term impact on quality of life. Together, these priorities will support the safe, effective, and culturally attuned integration of PAT into oncology care.

## 5. Conclusions

PAT shows considerable promise for alleviating psychological and existential suffering in PLWC. Our synthesis highlights both the efficacy and complexity of these interventions, underscoring the influence of therapeutic setting, mechanistic pathways, and clinical delivery models. Across these trials, it becomes clear that the experience of suffering is not merely a symptom to be eliminated, but often an opportunity for deeper insight, transformation, and renewal. As this field matures, oncology researchers and practitioners must lead the way in shaping evidence-based, patient-centered applications that bridge biological, psychological, and spiritual domains of healing.

## Figures and Tables

**Figure 1 curroncol-32-00380-f001:**
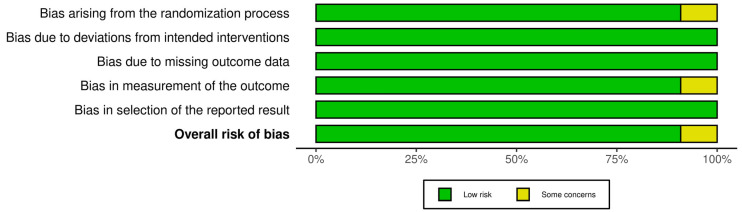
Risk of Bias assessment for included studies using the Cochrane RoB.2 tool under an Intention-to-Treat (ITT) analysis approach [21]. The summary plot displays risk assessments across five domains. Green indicates low risk, yellow indicates some concerns, and red indicates high risk.

**Figure 2 curroncol-32-00380-f002:**
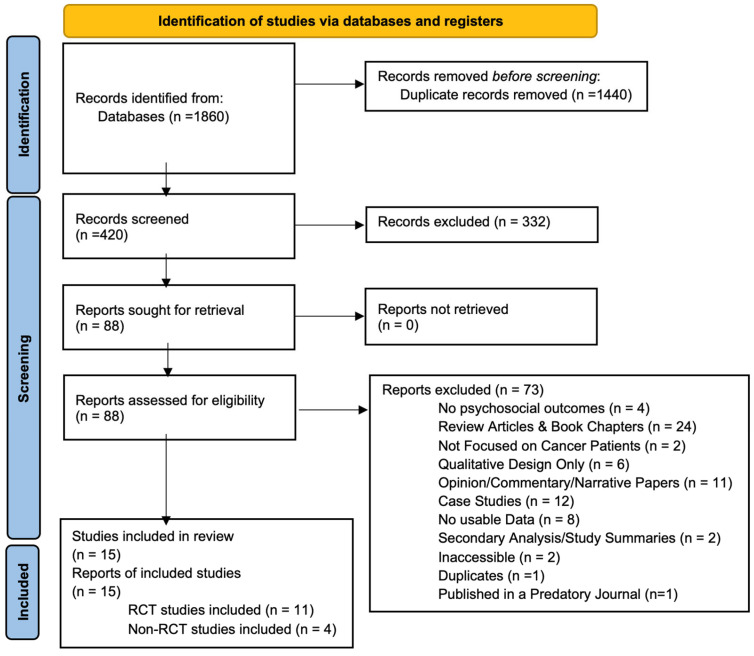
PRISMA 2020 flow diagram for citations assessing impact of psychedelics on patients with cancer or disease [20]. This is a flow diagram showing the initial citations discovered, the number of citations excluded after we applied our criteria, and the final number of studies included in the review.

**Figure 3 curroncol-32-00380-f003:**
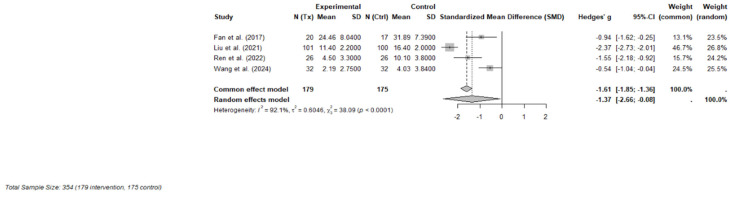
Forest plot of the effects of ketamine or esketamine on psychosocial symptoms in PLWC (random-effects model) [10,32,33,34]. This forest plot presents Hedges’ g for each included RCT comparing ketamine or esketamine to a control (saline or midazolam). The size of each square represents the weight of the study under the random-effects model. The horizontal lines indicate 95% confidence intervals. The blue diamond represents the pooled effect estimate. A negative effect size indicates greater reduction in symptoms for the ketamine group.

**Figure 4 curroncol-32-00380-f004:**
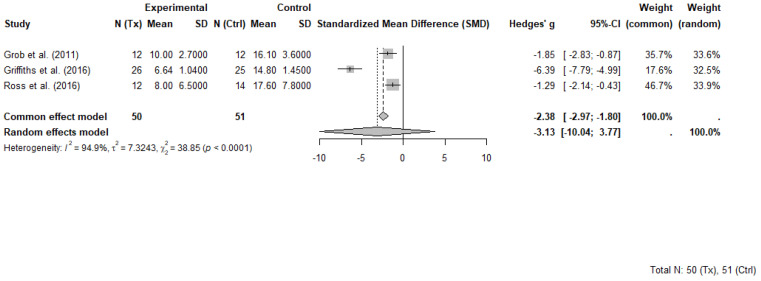
Forest plot of psilocybin-assisted therapy on psychosocial symptoms in people with cancer (random-effects model) [8,9,31]. Random-effects meta-analysis of psilocybin-assisted therapy on psychosocial symptoms in adults with cancer. Forest plot displaying Hedges’ g and 95% confidence intervals (CI) for three randomized controlled trials comparing psilocybin with placebo or active control conditions. Negative values indicate greater symptom reduction in the psilocybin group. Data are based on primary depression or psychological distress measures reported at the first post-treatment timepoint prior to crossover. Total sample included 50 participants in psilocybin groups and 51 in control groups.

**Table 1 curroncol-32-00380-t001:** Risk of Bias Assessment for Non-Randomized Studies Using the NIH Before-After (Pre-Post) Tool [22].

Criteria	Rosenblat et al., 2023 [12]	Shnayder et al., 2023 [23]	Lewis et al., 2023 [24]	Agrawal et al., 2024 [25]
Study objectives stated	Yes	Yes	Yes	Yes
Study population defined	Yes	Yes	Yes	Yes
Study participants representative of clinical populations of interest	Yes	Yes	Yes	Yes
All eligible participants enrolled	No	No	No	No
Sample size sufficient and/or described	No	No	No	No
Intervention clearly described	Yes	Yes	Yes	Yes
Outcome measures clearly described, valid, reliable	Yes	Yes	Yes	Yes
Blinding of outcome assessors	No	No	No	No
Follow-up rate	Yes	Yes	Yes	Yes
Statistical analysis	Yes	Yes	Yes	Yes
Multiple outcome measures	No	No	No	No
Group-level interventions and individual-level outcome efforts	N.A.	N.A.	N.A.	N.A.
Overall rating	Fair	Fair	Fair	Fair

Note: “Yes” indicates the criterion was met; “No” indicates the criterion was not met or was inadequately reported; “N.A.” criterion irrelevant to an individual-level intervention; *Fair* overall rating reflects at least one important limitation.

**Table 2 curroncol-32-00380-t002:** GRADE Summary of Evidence for the Effect of Ketamine and Psilocybin on Psychosocial Symptoms in Individuals with Cancer [30].

Outcome	No. of Studies	Study Design	Risk of Bias	Inconsistency	Indirectness	Imprecision	Overall Certainty	Effect Estimate (SMD [95% CI])
Reduction in psychosocial symptoms (e.g., depression, anxiety) with ketamine or esketamine vs. Control.	4	RCTs	Moderate ^1^	Serious ^2^	Not serious	Serious ^3^	Low	−1.37 [−2.66 to −0.08]
Reduction in psychosocial symptoms (e.g., depression, anxiety, existential distress) with psilocybin vs. Control.	3	RCTs	Moderate ^4^	Serious ^5^	Not serious	Serious ^6^	Low	−3.13 [−10.04 to 3.77]

^1^ Some concerns about risk of bias due to limited blinding and outcome measurement methods in individual trials. ^2^ Inconsistency is serious due to substantial heterogeneity (I^2^ = 92.1%) and variation in study populations and outcome measures. ^3^ Imprecision is serious because the 95% confidence interval is wide and includes effect sizes that range from potentially trivial to large. ^4^ Some concerns due to lack of blinding, crossover design, and reliance on self-report measures in small trials. ^5^ Serious inconsistencies due to very high heterogeneity (I^2^ = 94.9%, τ^2^ = 7.32), suggesting substantial variability in effect sizes. ^6^ Imprecision is serious due to wide 95% confidence interval that crosses both no effect and large effect, despite total sample size exceeding 100.

**Table 3 curroncol-32-00380-t003:** Overview of Included Randomized, Double-blind, Placebo-controlled Trials.

Study (Author, Year)	Sample Size (N)	Male (%)	Mean Age (SD)	Condition	Psychedelic Agent & Dose	Comparator	Therapy Included	Primary Outcome Measures	Evaluation Timepoints	Key Findings
Grob et al., 2011 [31]	12	8%	36–58	Advanced cancer-related anxiety	Psilocybin 0.2 mg/kg	Niacin (Vitamin B3)	Yes	BDI, POMS, STAI	6 months	Reduction in anxiety sustained at 6 months.
Griffiths et al., 2016 [8]	51	51%	56.3 (±1.4)	Cancer-related depression/ anxiety	Psilocybin 1 or 3 mg/70 kg (low) & 22 or 30 mg/70 kg (high)	Low-dose Psilocybin	Yes	GRID-HAM-D-17, HAM-A	6 months	Large effect size for depression & anxiety reduction (*p* < 0.001).
Ross et al., 2016 [9]	29	38%	56.28 (±12.93)	Cancer-related depression/ anxiety	Psilocybin 0.3 mg/kg	Niacin 250 mg	Yes	HADS, BDI, STAI	6.5 months	60–80% sustained reduction in anxiety & depression.
Fan et al., 2017 [32]	39	32%	45.78 (±14.4)	Cancer with suicidal ideation	Ketamine 0.5 mg/kg IV	Midazolam 0.05 mg/kg	No	BSI, MADRS-SI	Day 1, 3, 7	Significant reduction in suicidal ideation (*p* < 0.001).
Liu et al., 2020 [10]	303	0%	47.43 (±9.4)	Breast cancer and depression	S-Ketamine 0.125 mg/kg IV	Saline	No	HAMD-17, BDNF, 5-HT	3 days, 1 week, 1 month	Lower depression scores & higher BDNF & 5-HT levels.
Ren et al., 2022 [33]	104	51%	61.35 (±7.24)	Colorectal cancer surgery	Ketamine 0.1–0.3 mg/kg IV	Saline	No	HADS, QoR-40, IL-6, IL-8	24, 48, 72 h	Improvement in anxiety, depression, and inflammatory markers.
Wang et al., 2024 [34]	64	0%	42.05 (±5.81)	Breast cancer and depressive symptoms	Esketamine 0.2 mg/kg IV	Saline	No	PHQ-9, VAS	1, 3, 7, 30 days	PHQ-9 significantly lower at post-op day 1 (*p* = 0.047).
Gasser et al., 2014 [35]	12	50%	51.7 (±9.1)	Life-threatening illness with anxiety	LSD 200 µg	LSD 20 µg (active placebo)	Yes	STAI, EORTC-QLQ-30, HADS	2 months, 12 months	Large effect size for anxiety reduction (*p* = 0.033).
Holze et al., 2023 [36]	42	52%	45 (±12)	Anxiety (with/without life-threatening illness)	LSD 200 µg	Placebo	Yes	STAI-G, BDI, HAM-D-21	16 weeks	Significant reductions in anxiety (*p* = 0.007) & depression (*p* = 0.0004).
Wolfson et al., 2020 [37]	18	22.2%	54.9 (±7.9)	Life-threatening illness with anxiety	MDMA 125 mg	Placebo	Yes	STAI, BDI-II, PSQI	1 month, 2 months	Reduction in anxiety sustained at 2-month follow-up.
Lewis et al., 2023 [24]	12	33%	48.2 (±11.5)	Cancer patients with DSM-5 depressive disorder	Psilocybin 25 mg oral, group session	None (open-label, single-arm study)	Yes	HAMD-17	Baseline; 2 weeks; 26 weeks	HAM-D dropped from 21.5 to 10.1 (2 weeks) and 14.8 (26 weeks) (*p* < 0.001, *p* = 0.006).
Shnayder et al., 2023 [23]	30	30%	56 (±12)	Cancer patients with MDD	Psilocybin 25 mg oral	None (open-label, single-arm study)	Yes	NIH-HEALS	Baseline; 1, 3, 8 weeks	NIH-HEALS increased by ~16.4 points at 8 weeks (*p* < 0.001).
Agrawal et al., 2024 [25]	30	30%	56 (±12)	Curable/metastatic cancer with MDD	Psilocybin 25 mg oral, group session	None (open-label, single-arm study)	Yes	MADRS	Baseline; 1-week; 8 weeks	MADRS reduced by 19.1 points by Week 8 (*p* < 0.0001).
Rosenblat et al., 2023 [12]	20	35%	58.4 (±17.2)	Advanced cancer with MDD (palliative care)	Intranasal Ketamine was administered in three flexible doses (50–150 mg) over one week	None (open-label, single-arm study)	No	MADRS	Baseline; Day 8; Day 14	70% response, 45% remission; MADRS fell ~20 points by Day 8 (*p* < 0.001).

## Data Availability

No new data were created or analyzed in this study. All data used in the synthesis were obtained from previously published articles included in the systematic review and are publicly available through the original sources. A full list of included studies with citations is provided in the manuscript.

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
