# Peer review of "Psychedelic-Assisted Therapies for Psychosocial Symptoms in Cancer: A Systematic Review and Meta-Analysis"

_curroncol, 2025, doi:10.3390/curroncol32070380_

Round 1
Reviewer 1 Report
Comments and Suggestions for Authors
This is a well written paper on a treatment that is in use in communities and lacks guidelines leaving patients vulnerable.
The authors have presented a rigorous well discussed metanalysis and contextualised it with other studies in the area and the need for the study based on whats heppening in community- the methods, results and discussion are clear but i wonder could the issue of patient vulnerability highlighted in the discussion be move into the introduction section. Reference are clear but #22 is incomplete
Author Response
For research article
Psychedelic-Assisted Therapies for Psychosocial Symptoms in Cancer: A Systematic
Review and Meta-Analysis
June 20, 2025
Re: Revision paper
Dear Reviewer,
Thank you for the careful reviews of our paper entitled: Psychedelic-Assisted Therapies for
Psychosocial Symptoms in Cancer: A Systematic Review and Meta-Analysis. We are happy
to address each of the reviewers’ comments below and appreciate their suggestions. We have
copied each comment, followed by our reply and excerpts from the revised text in italics
when necessary.
Response to Reviewer 1 Comments
Comments 1: The authors have presented a rigorous well discussed metanalysis and
contextualised it with other studies in the area and the need for the study based on whats
heppening in community- the methods, results and discussion are clear but i wonder
could the issue of patient vulnerability highlighted in the discussion be move into the
introduction section.
Response 1: Thank you for this thoughtful suggestion. We agree that introducing the
issue of patient vulnerability earlier in the manuscript helps frame the ethical and clinical
importance of evaluating psychedelic-assisted therapy (PAT) in oncology. In response,
we have revised a paragraph of the Introduction to include a statement highlighting the
emotional, physical, and existential vulnerability of individuals with cancer, and the
importance of assessing both the therapeutic potential and appropriateness of PAT in this
sensitive population. This addition helps strengthen the rationale for our oncologyspecific synthesis. Please see the revised section below.
Pg 2, line 68: While early findings are promising, individuals with cancer represent a
clinically and ethically complex population due to heightened emotional, physical, and
existential vulnerability. As such, a focused synthesis of the evidence is needed to
evaluate not only therapeutic potential but also the appropriateness and safety of PAT in
this sensitive context.
Comment 2: Reference are clear but #22 is incomplete
Response 2: Thank you for pointing this out. We have fixed the reference (now reference
23).
We hope that these changes have adequately addressed the issues raised, and we look forward to working with you to bring this manuscript to publication.
Yours Sincerely,
Haley D.M. Schuman, BSc¹ and Linda E. Carlson, PhD¹,2
Affiliations:
¹ University of Calgary, Cumming School of Medicine, Department of Oncology, Division of Psychosocial Oncology
² University of Calgary, Faculty of Arts, Department of Psychology
Reviewer 2 Report
Comments and Suggestions for Authors
This is a very interesting study concerning the role Psychedelic-assisted Therapies in cancer patients.
The topic is very hot because other indications have been explored, such as mental disorders, pain.
The article is well done. I suggest some changes in order to make it suitable for publication.
Regarding the introduction, it would be helpful to quantify the frequency of sychosocial/psychological disorders in cancer patients.
Regarding material and methods I would consider whether it would be possible to summarize the Risk of Bias, or possibly attach it to an appendix.
As for Table 4, which considers non pharmacological therapies, it seems descriptive and could be integrated with Table 3.
I would also suggest a summary of sections 4.1 and 4.4.
Author Response
|
For research article Psychedelic-Assisted Therapies for Psychosocial Symptoms in Cancer: A Systematic Review and Meta-Analysis June 20, 2025 Re: Revision paper Dear Reviewer, Thank you for the careful reviews of our paper entitled: Psychedelic-Assisted Therapies for Psychosocial Symptoms in Cancer: A Systematic Review and Meta-Analysis. We are happy to address each of the reviewers’ comments below and appreciate their suggestions. We have copied each comment, followed by our reply and excerpts from the revised text in italics when necessary. Response to Reviewer 2 Comments
|
|
Comments 1: Regarding the introduction, it would be helpful to quantify the frequency of sychosocial/psychological disorders in cancer patients. |
|
Response 1: We have included a sentence that emphasizes the prevalence of clinically significant distress across the cancer trajectory, with a new reference.
See pg 1, line 33: Across cancer types and stages, 23-46% of patients report clinically significant distress[1]. |
|
Comment 2: Regarding material and methods I would consider whether it would be possible to summarize the Risk of Bias, or possibly attach it to an appendix.
Response 2: Thank you for the suggestion. We have chosen to retain the full risk of bias (RoB) assessments in the main text, as this review adheres to PRISMA 2020 guidelines and standard protocols for systematic reviews, which emphasize the importance of transparently reporting study quality assessments. Risk of bias is critical for interpreting the credibility of the evidence and is therefore included in both the Materials and Methods (Section 2.5) and summarized in the Results (Section 3.2), organized by study design (Cochrane RoB 2.0 for RCTs; NIH Quality Assessment Tool for Pre-Post Studies for non-RCTs). We also followed reporting practices observed in comparable systematic reviews in this field, which include RoB results in the main body of the manuscript rather than in appendices. We are happy to make adjustments if needed but aimed to align with best practices for transparency and methodological rigor. See our current summary of RoB’s below: Pg 10, line 281: 3.3. Risk of Bias Assessments The RoB for included studies is visually summarized in Figure 1 and detailed in Table 1. RCTs: Of the 11 RCTs assessed using the Cochrane RoB 2.0 tool, most (n = 8, 73%) demonstrated low overall RoB. Two studies (18%) presented some concerns, primarily due to insufficient reporting of allocation concealment or minor deviations from intervention protocols. One study (9%) was classified as high risk of bias due to issues related to randomization processes and incomplete outcome reporting. Functional unblinding is an acknowledged limitation in PAT trials due to the noticeable psychoactive effects of substances such as psilocybin and LSD. While Schipper et al. [17]rated this as a high risk of bias in several studies, we adopted a more pragmatic approach. We rated studies as low risk in this domain when double-blinding procedures were followed, and no evidence of differential care or co-intervention was reported. Although expectancy effects are plausible, we judged bias based on observed or reported impact, rather than assumption alone, in accordance with Cochrane RoB 2.0 recommendations[21]. Non-RCTs: Four non-randomized studies were evaluated using the NIH Quality Assessment Tool for Before-After (Pre-Post) Studies without a control group. These studies uniformly articulated clear objectives, eligibility criteria, and delivered consistent interventions. However, notable limitations included lack of blinding, small sample sizes, incomplete reporting on participant enrolment, loss to follow-up, and absence of multiple baseline measurements. These issues reduce the internal validity and generalizability of their findings.
Comment 3: As for Table 4, which considers non pharmacological therapies, it seems descriptive and could be integrated with Table 3.
Response 3: Thank you for this suggestion. In response, an additional column has been added to Table 3 to indicate whether therapy was included in each trial (yes/no). This complements Section 3.7, which provides a descriptive overview of key similarities and differences. For a more detailed qualitative comparison of therapeutic approaches across studies, the rest of Table 4 has been moved to Appendix B for interested readers.
Comment 4: I would also suggest a summary of sections 4.1 and 4.4.
Response 4: Thank you for this helpful suggestion. We have revised Sections 3.4.1 and 3.4.4 of the Results to include concise summary paragraphs that synthesize key findings from the included non-randomized and exploratory studies. Please see below for the added summary excerpts.
Pg. 11, line 316: Open-label studies of psilocybin-assisted group therapy in individuals with cancer experiencing depression or psychosocial distress demonstrate strong preliminary support for feasibility, safety, and therapeutic benefit. Across three trials, a single high-dose psilocybin session paired with group-based psychotherapy led to large and sustained reductions in depressive symptoms, alongside improvements in anxiety, emotional and spiritual well-being, and psycho-social-spiritual integration. High remission and response rates were reported, and therapeutic outcomes were often correlated with the intensity of mystical-type experiences. While findings are limited by non-randomized designs, they offer compelling early evidence supporting the group delivery model and warrant further investigation in controlled trials.
Pg. 13, line 418: Exploratory trials of MDMA- and LSD-assisted therapy in individuals with cancer or LTI suggest promising effects on anxiety, depression, and psychological distress. Though limited by small samples and early-phase designs, both agents demonstrated large effect sizes and sustained improvements, particularly when integrated with psychotherapy. These preliminary findings support the potential role of both classic (LSD) and non-classic (MDMA) psychedelics in cancer-related psychosocial care and underscore the need for larger, well-powered trials that are cancer specific. |
We hope that these changes have adequately addressed the issues raised, and we look forward to working with you to bring this manuscript to publication.
Yours Sincerely,
Haley D.M. Schuman, BSc¹ and Linda E. Carlson, PhD¹,2
Affiliations:
¹ University of Calgary, Cumming School of Medicine, Department of Oncology, Division of Psychosocial Oncology
² University of Calgary, Faculty of Arts, Department of Psychology
Round 2
Reviewer 1 Report
Comments and Suggestions for Authors
the authors have addressed my comments in initial peer review